# Heritability of the shape of subcortical brain structures in the general population

Gennady V. Roshchupkin[1,2,*], Boris A. Gutman[3,*], Meike W. Vernooij[1,4], Neda Jahanshad[3], Nicholas G. Martin[5], Albert Hofman[4,6], Katie L. McMahon[7], Sven J. van der Lee[4], Cornelia M. van Duijn[4,8], Greig I. de Zubicaray[9], André G. Uitterlinden[10], Margaret J. Wright[7,11], Wiro J. Niessen[1,2,12], Paul M. Thompson[3], M. Arfan Ikram[1,4,13,**] & Hieab H.H. Adams[1,4,**]

The volumes of subcortical brain structures are highly heritable, but genetic underpinnings of their shape remain relatively obscure. Here we determine the relative contribution of genetic factors to individual variation in the shape of seven bilateral subcortical structures: the nucleus accumbens, amygdala, caudate, hippocampus, pallidum, putamen and thalamus. In 3,686 unrelated individuals aged between 45 and 98 years, brain magnetic resonance imaging and genotyping was performed. The maximal heritability of shape varies from 32.7 to 53.3% across the subcortical structures. Genetic contributions to shape extend beyond influences on intracranial volume and the gross volume of the respective structure. The regional variance in heritability was related to the reliability of the measurements, but could not be accounted for by technical factors only. These findings could be replicated in an independent sample of 1,040 twins. Differences in genetic contributions within a single region reveal the value of refined brain maps to appreciate the genetic complexity of brain structures.

[1] Department of Radiology and Nuclear Medicine, Erasmus MC, Rotterdam 3015 CE, The Netherlands. [2] Department of Medical Informatics, Erasmus MC, Rotterdam 3015 CE, The Netherlands. [3] Imaging Genetics Center, Mark and Mary Stevens Neuroimaging and Informatics Institute, Keck School of Medicine of the University of Southern California, Marina del ReyLos Angeles, California 90292, USA. [4] Department of Epidemiology, Erasmus MC, Rotterdam 3015 CE, The Netherlands. [5] QIMR Berghofer Medical Research Institute, Brisbane, Queensland 4006, Australia. [6] Department of Epidemiology, Harvard T.H. Chan School of Public Health, Boston, Massachusetts 02115, USA. [7] Centre for Advanced Imaging, The University of Queensland, Brisbane, Queensland 4072, Australia. [8] Translational Epidemiology, Faculty Science, Leiden University, Leiden, 2333 CC, The Netherlands. [9] Faculty of Health, Institute of Health and Biomedical Innovation, Queensland University of Technology, Brisbane, Queensland 4059, Australia. [10] Department of Internal Medicine, Erasmus MC, Rotterdam 3015 CE, The Netherlands. [11] Queensland Brain Institute, The University of Queensland, Brisbane, Queensland 4072, Australia. [12] Faculty of Applied Sciences, Delft University of Technology, Delft 2628 CJ, The Netherlands. [13] Department of Neurology, Erasmus MC, Rotterdam 3015 CE, The Netherlands. * These authors contributed equally to this work. ** These authors jointly supervised this work. Correspondence and requests for materials should be addressed to M.A.I. (email: m.a.ikram@erasmusmc.nl).

Subcortical brain regions are important for a multitude of biological processes, including cognitive and motor functions[1,2]. There is substantial structural variation in these regions, both within the normal range[3] and in the context of various neuropsychiatric diseases[4,5]. Factors driving individual variation could provide insight into brain development, healthy ageing and pathological states, but these remain largely unknown. Variation in subcortical brain structures is affected by environmental factors, such as education, diet and stress, but a considerable proportion of the variation is determined by genes[6,7]. A recent twin study of gross subcortical volumes found heritability estimates ranging between 0.44 and 0.88 (ref. 8), which were especially high for the caudate and thalamus.

Even so, aggregate measures such as volume do not capture the complexity of subcortical structures. The hippocampus, for example, is made up of several subfields, each with partially independent functional roles. More recently, image processing methods have been developed to characterize brain structure beyond purely volumetric measures, and yielding a range of shape descriptors[9–13]. The high-dimensionality allows the detection of more localized differences in brain structure, and shape can provide relevant biological information in addition to aggregate measures[14–17]. Several genetic variants that influence the volume of subcortical structures have been identified[18–20], but their effect could be localized to certain sub-regions using shape analyses[19,20]. However, the extent to which genes contribute to the variability in shape of subcortical structures has yet to be determined.

Here we quantify genetic influences on shape variability of 14 subcortical brain structures in 3,686 unrelated individuals from the population-based Rotterdam Study. We compare the heritability of vertex-wise shape measures to gross volumes as well as other aggregate measures of shape obtained through dimension-reduction techniques. We show that the shape of subcortical structures is under genetic control, and investigate the relation of the resulting profiles with the gross volume and measures of reproducibility.

## Results

**Study population.** The characteristics of the study population are shown in Table 1. The mean age of the Rotterdam study population was $65.9 \pm 10.9$ years, and 55.0% were women. For the 14 subcortical structures, the mean volumes were between 0.49 and 6.25 ml. For the Queensland Twin IMaging (QTIM) study, mean age was $22.9 \pm 2.8$ years, and 61.6% were women. Mean subcortical volumes were higher than in the Rotterdam study across the board, ranging from 0.79 and 7.82 ml.

**Heritability of volume and shape of subcortical structures.** The structure of subcortical brain regions was quantified by calculating their gross volume as well as two measures of their shape. Age- and sex-adjusted heritability estimates for the gross volume of each of the subcortical structures were between 1.6 and 43.4% (Table 2). For the two vertex-wise shape measures, the maximal heritability estimates per structure ranged from 32.7 to 53.3% (Table 2). Both the radial distance (Fig. 1a–c) and the Jacobian determinant (Fig. 1d–f) showed clusters of high heritability under various models. Further adjustment for intracranial volume did not influence results (Fig. 1), and estimates were highly correlated between both models (Supplementary Fig. 1). The addition of the structure-specific gross volume to the model, however, did affect the heritability distribution across the structures (Fig. 1), particularly for the shape measures that are highly correlated with the gross volume (Supplementary Fig. 2).

**Table 1 | Characteristics of the study population.**

| Characteristic | Rotterdam Study (N = 3,686) | QTIM (N = 1,040) |
|---|---|---|
| Age, mean (s.d.), years | 65.9 (10.9) | 22.9 (2.8) |
| Female sex, n (%) | 2,029 (55.0%) | 641 (61.6%) |
| Intracranial volume, mean (s.d.), cm³ | 1,478.6 (161.3) | 1,484 (157.1) |
| *Left hemisphere, mean (s.d.), cm³* | | |
| Accumbens | 0.56 (0.10) | 0.83 (0.15) |
| Amygdala | 1.31 (0.21) | 1.84 (0.25) |
| Caudate | 3.40 (0.56) | 3.76 (0.50) |
| Hippocampus | 3.84 (0.62) | 4.32 (0.46) |
| Pallidum | 1.47 (0.24) | 1.61 (0.25) |
| Putamen | 4.62 (0.68) | 6.60 (0.72) |
| Thalamus | 6.25 (0.79) | 7.82 (0.89) |
| | | |
| *Right hemisphere, mean (s.d.), cm³* | | |
| Accumbens | 0.49 (0.09) | 0.79 (0.11) |
| Amygdala | 1.39 (0.22) | 1.88 (0.25) |
| Caudate | 3.51 (0.58) | 3.92 (0.53) |
| Hippocampus | 3.85 (0.59) | 4.32 (0.46) |
| Pallidum | 1.41 (0.25) | 1.53 (0.18) |
| Putamen | 4.45 (0.65) | 6.00 (0.65) |
| Thalamus | 6.25 (0.79) | 7.43 (0.88) |

QTIM, Queensland Twin Imaging; SD, standard deviation.

**Reproducibility of subcortical shape.** Next, we investigated the relation between our heritability estimates and the reproducibility of subcortical shape. In a subset of 83 persons who were scanned twice within 1–9 weeks, we quantified the reproducibility by calculating intraclass correlation coefficients for the vertex-wise shape measures (Supplementary Fig. 3). There was considerable overlap between heritability and reproducibility (Fig. 2a,b), and both were correlated within hemisphere (Fig. 2c,d). Poorly reproducible shape measures were generally not heritable, whereas high reproducibility included the full range of heritability estimates (Fig. 2c,d).

**Heritability of shape measures through data reduction.** Finally, we explored whether high-dimensional shape data could be reduced to a smaller set of variables with a larger genetic contribution. We performed principal component analyses on the two vertex-wise shape measures for each structure and computed the heritability of the resulting components. Except for the Jacobian determinant of both hippocampi, the maximal heritability was lower than for the vertex-wise measures (Table 2). Similarly, the components were in general less heritable than the vertex-wise measures (Fig. 3). Furthermore, the order of the components based on the eigenvalues did not correlate well with the order based on the heritability ($\rho$ ranges from $-0.038$ to 0.096; Supplementary Table 1).

**Replication of heritability in twins.** The maximum heritability estimates for the two vertex-wise shape measures per structure ranged from 48.9 to 78.3%. Both the radial distance (Supplementary Fig. 3A–C) and the Jacobian determinant (Supplementary Fig. 4D–F) showed clusters of high heritability under various models. Further adjustment for intracranial volume did not influence the results (Supplementary Fig. 4C,E). The addition of the structure-specific gross volume to the model, however, did affect the heritability distribution across the structures (Supplementary Fig. 4C,F). Comparing the results of the twin-based and population study, we found a considerable

**Table 2 | Heritability estimates of various structural measures of subcortical brain regions.**

| Region | Gross volume | | Radial distance | | Jacobian determinant | | PCA radial distance | | PCA Jacobian determinant | |
|---|---|---|---|---|---|---|---|---|---|---|
| | $h^2$ | P | $h^{2\star}$ | P | $h^{2\star}$ | P | $h^{2\star}$ | P | $h^{2\star}$ | P |
| *Left hemisphere* | | | | | | | | | | |
| Amygdala | 8.1 | 0.18 | 47.7 | $1.72 \times 10^{-6}$ | 35.4 | $2.85 \times 10^{-4}$ | 29.9 | $4.40 \times 10^{-4}$ | 27.9 | $9.30 \times 10^{-4}$ |
| Accumbens | 11.6 | 0.099 | 34.0 | $4.71 \times 10^{-4}$ | 33.7 | $5.11 \times 10^{-4}$ | 28.7 | $7.04 \times 10^{-4}$ | 42.0 | $1.45 \times 10^{-6}$ |
| Caudate | 33.7 | $8.6 \times 10^{-5}$ | 49.9 | $6.33 \times 10^{-7}$ | 52.9 | $1.40 \times 10^{-7}$ | 42.4 | $1.20 \times 10^{-6}$ | 35.1 | $4.73 \times 10^{-5}$ |
| Hippocampus | 10.8 | 0.12 | 32.7 | $7.32 \times 10^{-4}$ | 29.2 | $2.23 \times 10^{-3}$ | 28.9 | $6.59 \times 10^{-4}$ | 29.6 | $5.03 \times 10^{-4}$ |
| Pallidum | 32.2 | $1.7 \times 10^{-4}$ | 39.6 | $5.75 \times 10^{-5}$ | 44.1 | $8.65 \times 10^{-6}$ | 30.8 | $2.96 \times 10^{-4}$ | 27.0 | $1.33 \times 10^{-3}$ |
| Putamen | 43.4 | $6.8 \times 10^{-7}$ | 49.4 | $7.43 \times 10^{-7}$ | 52.7 | $1.45 \times 10^{-7}$ | 34.1 | $7.16 \times 10^{-5}$ | 40.7 | $2.92 \times 10^{-6}$ |
| Thalamus | 34.1 | $7.4 \times 10^{-5}$ | 53.3 | $1.05 \times 10^{-7}$ | 45.3 | $5.07 \times 10^{-6}$ | 30.2 | $3.78 \times 10^{-4}$ | 29.4 | $5.26 \times 10^{-4}$ |
| *Right hemisphere* | | | | | | | | | | |
| Amygdala | 20.4 | 0.012 | 33.5 | $5.45 \times 10^{-4}$ | 31.5 | $1.08 \times 10^{-3}$ | 30.5 | $3.45 \times 10^{-4}$ | 27.7 | $1.03 \times 10^{-3}$ |
| Accumbens | 1.6 | 0.43 | 33.1 | $6.30 \times 10^{-4}$ | 35.1 | $3.13 \times 10^{-4}$ | 34.5 | $5.99 \times 10^{-5}$ | 31.7 | $2.10 \times 10^{-4}$ |
| Caudate | 34.7 | $5.4 \times 10^{-5}$ | 46.7 | $2.86 \times 10^{-6}$ | 47.5 | $1.95 \times 10^{-6}$ | 29.9 | $4.45 \times 10^{-4}$ | 33.8 | $8.75 \times 10^{-5}$ |
| Hippocampus | 8.0 | 0.19 | 33.7 | $5.26 \times 10^{-4}$ | 17.7 | $4.23 \times 10^{-2}$ | 30.8 | $3.00 \times 10^{-4}$ | 28.9 | $6.44 \times 10^{-4}$ |
| Pallidum | 36.6 | $2.3 \times 10^{-5}$ | 46.4 | $3.12 \times 10^{-6}$ | 44.5 | $7.22 \times 10^{-6}$ | 41.4 | $1.97 \times 10^{-6}$ | 29.2 | $5.77 \times 10^{-4}$ |
| Putamen | 37.1 | $1.8 \times 10^{-5}$ | 42.6 | $1.70 \times 10^{-5}$ | 37.5 | $1.32 \times 10^{-4}$ | 32.7 | $1.36 \times 10^{-4}$ | 33.4 | $1.01 \times 10^{-4}$ |
| Thalamus | 30.8 | $3.0 \times 10^{-4}$ | 46.2 | $3.50 \times 10^{-6}$ | 50.4 | $4.50 \times 10^{-7}$ | 37.1 | $1.78 \times 10^{-5}$ | 31.8 | $2.02 \times 10^{-4}$ |

$h^2$, heritability estimate in %; PCA, principal component analysis.
*Estimate indicates highest heritability among all vertices or principal components.

Basic model: age-and sex-adjusted — Basic model plus intracranial volume — Basic model plus structure volume

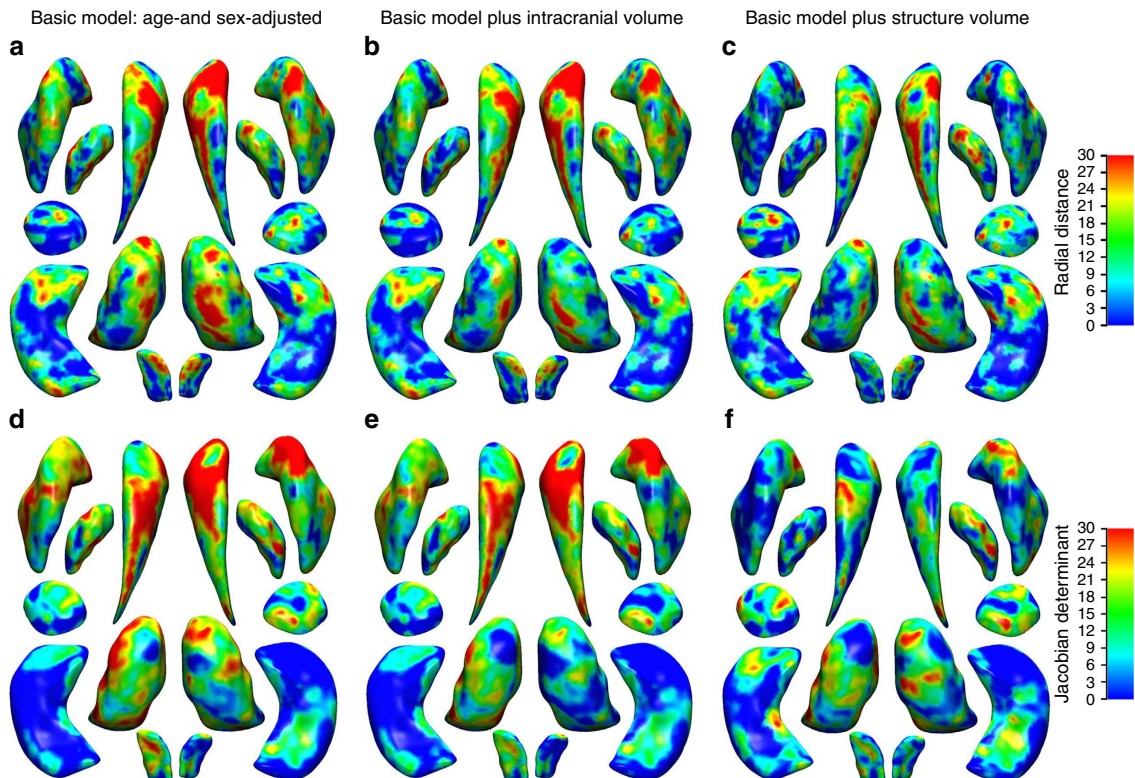

**Figure 1 | Heritability maps of shape measures of subcortical brain regions under various models.** Maps show the heritability of seven bilateral subcortical structures for the shape measures of radial distance (**a–c**) and the Jacobian determinant (**d–f**). Heritability estimates were obtained using three different statistical models: a basic model with age and sex (**a,d**), and additionally adjusting for either intracranial volume (**b,e**) or the volume of the specific structure (**c,f**).

overlap and significant correlation ($P$ value $= 3.03 \times 10^{-306}$) in estimated heritability (Supplementary Fig. 5).

## Discussion

Here we show that, in a general population of middle-aged and elderly individuals, the shapes of subcortical structures are under genetic control. The vertex-wise heritability is higher than for aggregate measures such as volume and principal components. Moreover, the heritability pattern underlines the importance of reproducibility in deriving shape measures, but also reveals that the extent of genetic influences is not uniformly distributed across subcortical structures. We confirmed our findings in an independent cohort of twins, suggesting that the genetic

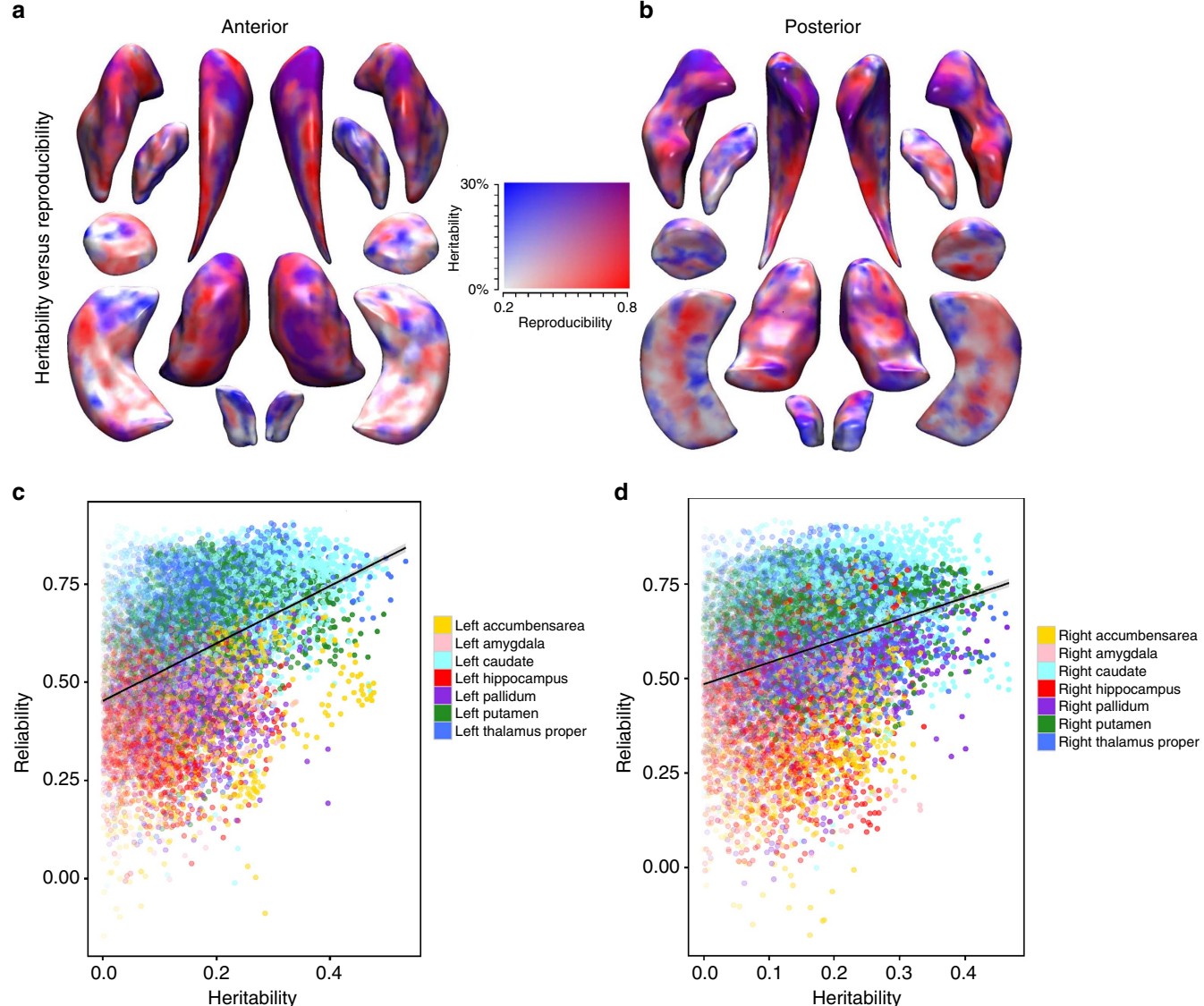

**Figure 2 | Concordance between the heritability of subcortical shape and reproducibility of the measures.** Figure showing the concordance between the heritability of the shape (radial distance) of subcortical structures and the reproducibility of these measures. Maps illustrate heritability (high is red) and reproducibility (high is blue) and their overlap (purple) from the anterior (**a**) and posterior (**b**) direction. Scatter plots between heritability and reprodcubility of the left (**c**) and right (**d**) hemisphere for the seven subcortical structures. Colours indicate the different structures (see figure legends).

architecture of subcortical shapes is similar across populations, despite differences in the sample, the study design, scanner types and methods to compute the heritability.

The higher vertex-wise heritability could reflect true biological differences in the degree of genetic contribution to the variability in shape. For the cerebral cortex, it has already been shown that different genes influence distinct parts of the brain and that the heritability also differs between regions[21–23]. Subcortical structures are also heterogeneous and consist of functionally diverging sub-regions, such as the nuclei of the pallidum or the head and tail of the caudate. Our results are in line with a recent study by Whelan *et al.*[24] showing that hippocampal subfields differ in their heritability. However, methodological reasons for this difference in heritability should also be considered. Particularly, a lower signal-to-noise ratio in some of the measures might have influenced the results, leading to low heritability estimates. Issues in the segmentation or registration steps will thus obscure true biological differences if these systematically affect certain sub-regions of a structure. We

investigated whether this plays a role by overlapping our heritability maps with maps of the technical reproducibility. Indeed, shape measures that could be poorly reproduced were not heritable. However, while high reproducibility was required for detecting a substantial genetic component, it did not necessarily translate into a high heritability. For example, for the shape measures with a high reproducibility (intraclass correlation coefficients > 0.75), a wide range of heritability estimates was observed (0–53%). Thus, even when the signal-to-noise ratio was comparable, we still observed regional differences in the degree of genetic contribution. The highly heritable measures are interesting targets for more in-depth genetic studies.

Heritability estimates calculated in our analysis represent both upper and low bounds of narrow-sense heritability. Our results are consistent with the theory that twin-based heritability tends to be higher than population-based estimates. However, we did not find a high correlation between the results, which could be due to several factors. Our population study consisted of relatively older individuals, which may impact the heritability: the effects of non-

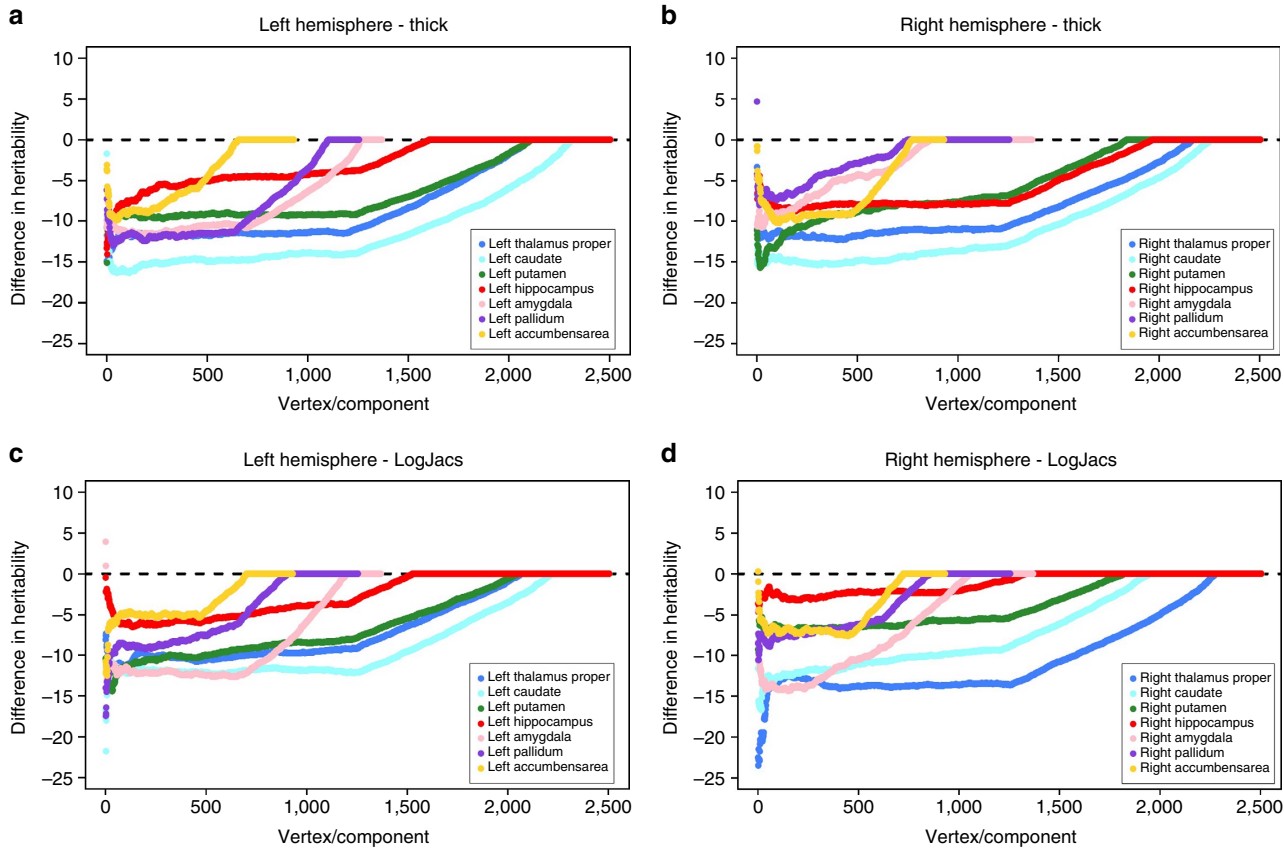

**Figure 3 | Difference in heritability between vertex-wise shape measures and PCA components.** Plots illustrate the difference between heritability estimates obtained from the vertex-wise shape measures and the heritability of the components obtained through principal component analysis for seven bilateral subcortical structures. Separate panels are provided for the shape measures of radial distance (**a,b**) and the Jacobian determinant (**c,d**) and the left (**a,c**) and right (**b,d**) hemisphere. All vertex-wise shape measures and principal components were first sorted in descending order of heritability, and the vertex-wise measures were substracted from the corresponding component's heritability. Colours indicate the different structures (see figure legends).

genetic factors on subcortical structures (for example, lifestyle factors) accumulate over an individual's lifetime and the overall contribution of genes might be reduced compared to younger individuals. Causal variants not captured on the genotyping array or through subsequent imputation also could lead to a different distribution of the heritability. In addition, apart from array limitations, non-additive genetic factors are not taken into account when computing population based heritability. These factors should be taken into account when interpreting our results.

An important question for future research on shape is which variables need to be controlled for in a regression analysis. Here we aimed to provide an answer by studying two controversial adjustment variables: the total intracranial volume and the gross volume of the structure under study. For the heritability estimates of shape, adjustment for intracranial volume did not affect the results, suggesting that the genes regulating shape are not general brain growth genes, but rather more specific for a structure or its sub-regions. The volume adjustments did change some of the results, but more so for vertices whose shape measures correlate most with the gross volume of the structure. Likely, the genes underlying a structure's gross volume are largely driven by these vertices as they typically represent the widest parts of a structure (highest mean radial distance), where radial measures tend to be highly correlated with its volume. Our results are in agreement with previous work[25], where the heritability of region-specific measures was reduced after adjustments for the total cortical surface area and thickness.

The detailed information provided by shape measures being their most attractive feature, the increase in dimensionality is potentially counterproductive, especially in the case of genetic homogeneity across a structure. We therefore also performed principal component analyses to demonstrate that the amount of variability explained by the components did not seem related to the heritability: near-zero correlations were found between the order of the components based on the eigenvalues and the heritability estimates. Although the principal component analysis captures most of the variation using fewer variables, methods, which are based on the genetic correlation, may lead to biologically more meaningful results.

While heritability provides an estimate of how much of the variance is determined by genetics, it does not point to specific genetic loci. The most commonly accepted method for gene discovery is to perform an unbiased screen of all genetic variants, that is, genome-wide association study (GWAS) to identify specific genetic factors. However, such efforts require large-scale collaborations in the order of tens of thousands of individuals to identify a robust association[18–20,26]. Furthermore, additional multiple testing correction should be considered when performing GWAS of 54,000 shape measures. This could lead to a loss of power if the effects are homogeneous across a structure. However, if the effects are localized and mostly affect specific vertices, then a GWAS of shape measures may actually increase power since the effect sized will be larger compared with a GWAS of an aggregate volume.

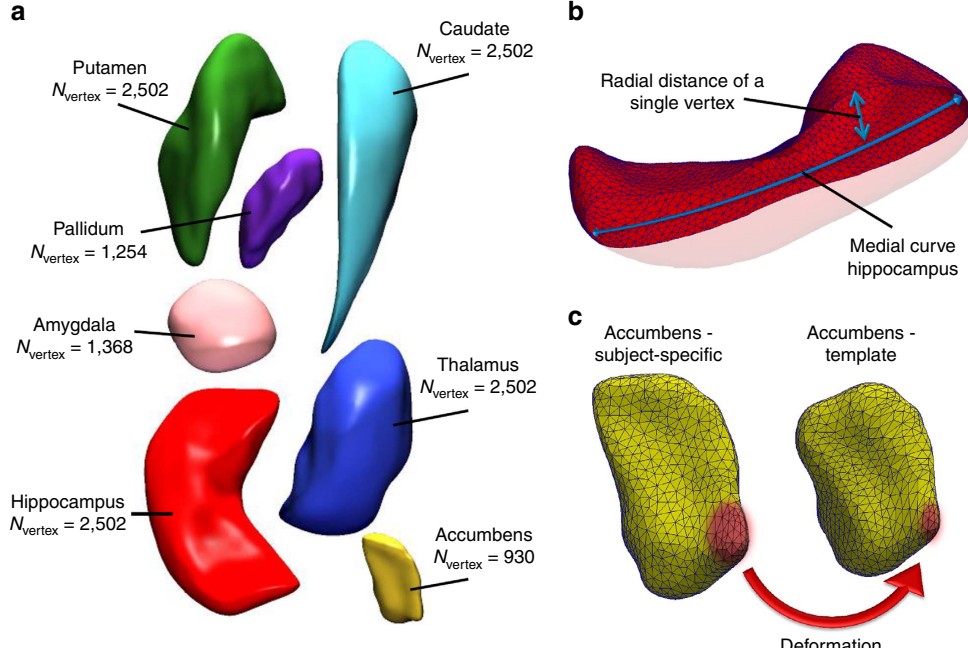

**Figure 4 | Subcortical brain structures and the derivation of shape measures.** Overview of the subcortical brain structures studied in this manuscript and the derivation of the shape measures. (**a**) The seven structures with corresponding number of vertices: accumbens, amygdala, caudate, hippocampus, pallidum, putamen and thalamus. (**b,c**) The two vertex-wise measures of shape: the radial distance is defined as the distance of a vertex to the medial curve of the structure, for example, the hippocampus in **b**. The Jacobian determinant captures the deformation that is needed to map a subject-specific shape to a template, which is shown with an example of the accumbens in **c**.

Data reduction methods always rely on assumptions and are often aimed at resolving computational issues. However, with the advent of big data collection, methods have been developed to analyse such large data sets efficiently. Software packages designed for high-dimensional data include MEGHA[27], for heritability analyses, BOLT-LMM[28], for genetic correlation analyses and HASE[29], for genome-wide association studies. These improvements in software, and also hardware, now pave the way for full-scale analyses without reliance on data reduction methods.

In conclusion, our work demonstrates that the shape of subcortical brain structures is a relevant phenotype for genetic studies, complementary to aggregated measures. Fine-scale maps of genetic influences on the brain are likely to reveal a complex mosaic of genetic modules, with partially divergent sets of genes that drive them.

## Methods

**Study population.** This work was performed in the Rotterdam Study[30], a population-based cohort study in the Netherlands including a total of 14,926 participants (aged 45 years or over at enrolment). The overall aim of the study is to investigate causes and determinants of chronic diseases in elderly people, the participants were not selected for the presence of diseases or risk factors. Since 2005, all participants underwent brain magnetic resonance imaging (MRI) to examine the causes and consequences of age-related brain changes[31]. Between 2005 and 2013, a total of 5,691 unique persons were scanned. The Rotterdam Study has been approved by the Medical Ethics Committee of the Erasmus MC and by the Ministry of Health, Welfare and Sport of the Netherlands, implementing the Wet Bevolkingsonderzoek: ERGO (Population Studies Act: Rotterdam Study). All participants provided written informed consent to participate in the study and to obtain information from their treating physicians.

Replication was performed in 1,040 healthy young adult twins from the QTIM project[32]. All participants of the imaging sample were Caucasian and right-handed for throwing and writing (Annett's Handedness Questionnaire). The genetic analyses were conducted in the 350 complete twin pairs ($n = 700$): 148 monozygotic (100 male), 120 dizygotic (39 male), and 82 opposite-sex pairs. Self-reported data were used to screen participants for contraindications for imaging as well as any significant medical, psychiatric or neurological conditions, history of substance abuse and current use of psychoactive medication. The study was approved by the Human Research Ethics Committees of the Queensland Institute of Medical Research, the University of Queensland, and Uniting Health Care, Wesley Hospital. Informed consent was obtained from each participant and parent or guardian for participants under 18 years of age.

**Genotyping and imputation.** Genotyping in the Rotterdam Study was performed using the Illumina 550 and 550K duo arrays[30]. Subsequently, we removed samples with call rate below 97.5%, gender mismatch, excess autosomal heterozygosity, duplicates or family relations and ancestry outliers, and variants with call rate below 95.0%, failing missingness test, Hardy–Weinberg equilibrium $P$ value $< 10^{-6}$, and minor allele frequency $< 1\%$. Genotypes were imputed using MACH/minimac software[33] to the 1000 Genomes phase I version 3 reference panel (all population).

For QTIM, genotyping of nine markers was used to determine the zygosity of same-sex twins, which was later confirmed for $> 92\%$ of the sample with the Illumina 610K SNP array.

**Image acquisition.** For the Rotterdam Study, MRI scanning was done on a 1.5-T MRI unit with a dedicated eight-channel head coil (GE Healthcare). The MRI protocol consisted of several high-resolution axial sequences, including a T1-weighted sequence (slice thickness 0.8 mm), which was used for further image processing. In addition, 85 persons were rescanned within days to weeks after the first scan to estimate the reproducibility of imaging-derived measures. A detailed description of the MRI protocol was presented by Ikram et al.[31]

The twin pairs of QTIM were scanned on a 4T Bruker Medspec (Bruker, Germany) whole body MRI system paired with a transverse electromagnetic (TEM) head coil. Structural T1-weighted three-dimensional images were acquired TR = 1,500 ms, TE = 3.35 ms, TI = 700 ms, 240 mm, field of view, 0.9 mm slice thickness, 256 or 240 slices depending on acquisition orientation (86% coronal (256 slices), 14% sagittal (240 slices)).

**Image processing.** The T1-weighted MRI scans were processed using FreeSurfer[34] (version 5.1) to obtain segmentations and volumetric summaries of the following seven subcortical structures for each hemisphere: nucleus accumbens, amygdala, caudate, hippocampus, pallidum, putamen and thalamus (Fig. 4a).

Next, segmentations were processed using a previously described shape analysis pipeline[9,10]. Briefly, a mesh model was created for the boundary of each structure. Subcortical shapes were registered using the 'Medial Demons' framework, which matches shape curvatures and medial features to a pre-computed template[35]. To do this, a medial model of each individual surface model is fit following Gutman et al.[36], and medial as well as intrinsic features of the shape drive registration to a template parametrically on the sphere. To minimize metric

distortion, the registration was performed in the fast spherical demons framework[10]. The templates and mean medial curves were previously constructed and are distributed as part of the ENIGMA-Shape package (http://enigma.usc.edu/ongoing/enigma-shape-analysis/).

The resulting meshes for the 14 structures consist of a total of 27,120 vertices (Fig. 4a). For these vertices, two measures were used to quantify shape: the radial distance and the natural logarithm of the Jacobian determinant. The radial distance represents the distance of the vertex from the medial curve of the structure (Fig. 4b). The Jacobian determinant captures the deformation required to map the subject-specific vertex to a template and indicates surface dilation due to subregional volume change (Fig. 4c). Detailed information is provided in the Supplementary Material.

Finally, we performed 28 principal component analyses: for each of the 14 subcortical structures and for both types of shape measures (radial distance and Jacobian determinant), we computed the full set of components. This yielded the same number of principal components as the original number of vertices that were used to describe shape (Fig. 4a). The components were sorted in descending order of the eigenvalues, which corresponds to the amount of explained variance of shape.

**Heritability estimation.** We used Massively Expedited Genome-wide Heritability Analysis (MEGHA)[28] to estimate heritability in our sample of unrelated individuals. This method allows fast and accurate estimates of heritability across thousands of phenotypes based on genome-wide genotype data of common genetic variants from unrelated individuals. As previously described[37], a genetic relationship matrix was constructed using the 1000 Genomes imputed genotypes, filtered on imputation quality ($R^2 < 0.5$) and allele frequency (MAF < 0.01). We calculated pairwise genetic relatedness between all individuals. We removed one person for pairs with more than 0.025 genotype similarity, resulting in a final study population of 3,686 subjects.

Twin-based heritability was estimated using maximum-likelihood variance components methods implemented in the SOLAR software (www.solar-eclipse-genetics.org, version 6.6.2)[38]. To test the hypothesis that no variance can be explained genetically, log likelihoods of models with no genetic components were compared with those with genetic and environmental components. As twice the log likelihood is distributed as a mixture of chi-squared distributions, the hypothesis test and $P$ value can be derived parametrically[38].

To correct for multiple comparisons across all vertices and all structures, we used the standard false discovery rate (FDR) threshold at $q = 0.05$ to localize regions of significant heritability within each of the subcortical structures[39].

**Data availability.** The data that support the findings of this study are available from the corresponding author upon reasonable request.

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

## Acknowledgements

The generation and management of GWAS genotype data for the Rotterdam Study are supported by the Netherlands Organization of Scientific Research NWO Investments (no. 175.010.2005.011, 911-03-012). This study is funded by the Research Institute for Diseases in the Elderly (014-93-015; RIDE2), the Netherlands Genomics Initiative (NGI)/Netherlands Organization for Scientific Research (NWO) project no. 050-060-810. The Rotterdam Study is funded by Erasmus Medical Center and Erasmus University, Rotterdam, Netherlands Organization for the Health Research and Development(ZonMw), the Research Institute for Diseases in the Elderly (RIDE), the Ministry of Education, Culture and Science, the Ministry for Health, Welfare and Sports, the European Commission (DG XII), and the Municipality of Rotterdam. This research is supported by the Dutch Technology Foundation STW (12723), which is part of the NWO, and which is

partly funded by the Ministry of Economic Affairs. This project has received funding from the European Research Council (ERC) under the European Union's Horizon 2020 research and innovation programme (project: ORACLE, grant agreement No: 678543). Further support was obtained through the Joint Programme—Neurodegenerative Disease Research working groups on High-Dimensional Research in Alzheimer's Disease (ZonMW grant number 733051031) and Full exploitation of High Dimensionality (ZonMW grant number733051032). This study was supported in part by 2014 NIH Big Data to Knowledge (BD2K) Initiative under U54EB020403. Additional funding was provided by the Michael J. Fox Foundation and Alzheimer's Association 'Biomarkers Across Neurodegenerative Diseases' (BAND) fellowship. QTIM was funded by the Australian National Health and Medical Research Council (project grants no. 496682 and 1009064) and US National Institute of Child Health and Human Development (RO1HD050735). We are grateful to the twins for their generosity of time and willingness to participate in our study. We also thank the many research assistants, radiographers, and other staff at QIMR Berghofer Medical Research Institute and the Centre for Advanced Imaging, University of Queensland.

## Author contributions

G.V.R. and B.A.G. jointly conceived the study, participated in its design, performed the analysis, interpreted the data and drafted the manuscript. M.W.V., N.J., N.G.M., A.H., K.L.M., S.J.v.d.L., C.M.v.D., G.I.d.Z., A.G.U., M.J.W., W.J.N. and P.M.T. acquired data and revised the manuscript critically for important intellectual content. H.H.H.A. and M.A.I. participated in its design, interpreted the data and revised the manuscript critically for important intellectual content. All authors read, edited and approved the manuscript.

## Additional information

**Competing financial interests:** W.J. Niessen is co-founder and shareholder of Quantib BV. The remaining authors declare no competing financial interests.

