## [Peer Review File · Nature Communications]

Reviewers' comments:

Reviewer #1 (Remarks to the Author):

The manuscript by Roshchupkin and colleagues provides the first estimate of the heritability of the shape of subcortical structures. The volumes of subcortical structures are known to be highly heritable and therefore one would expect the high heritability of the shape as well. However, this manuscript shows that the regional pattern of heritability differs quite a bit and may not be related to the methodological error.

This manuscript is highly novel. The manuscript uses new methodological approaches for both shape and genetic analyses. This may also raise the criticism since the variance in regional heritability for the structures remains unexplained. This manuscript could be greatly strengthened if authors perform a replication study in the related individual. A dataset collected by human connectome project offers the same spatial resolution and heritability. It is collected in the related individuals and therefore heritability measurements could be calculated directly. I suggest repeating these analyses in the HCP dataset to show that the regional variance in heritability are indeed present and independent of the cohort or methodological issues.

Reviewer #2 (Remarks to the Author):

This study is the first to show that individual differences in the shape of subcortical brain structures is to an important extent heritable.

The study is novel and original in the field of brain studies by focusing on shape [instead of volume, white matter etc], and by using genetic data of unrelated individuals [instead of twin or family data].

Analyses are well performed, and additional analyses on reliability and data reduction provide added value to the interpretation and robustness of the results.

Yet, I have some questions related to the study.

Data:

The sample used is the Rotterdam Study that aims to study 'causes and determinants of chronic diseases in elderly' which raises the question whether the prevalence of disease is higher compared to the general population, and if yes, whether this would affect the results.

In addition the inclusion of subjects raises some questions:

1. On which basis were subjects included for scanning and genotyping (~35% of total sample)?
2. After genetic QC, 3868 subjects remained in the study (page 7, but Table 1 states 3686?). After imaging QC no subjects were excluded?

Statistics

The analyses and results are well presented, apart from some of the Figures that need clarification: I miss Figure Legends in Supplementary Figures 1 and 2. E.g., text x-axis is too small, and thus hard to read. In addition, Figure 4 is hard to read for me. What is meant with 'measure number' on the x-axis? What is the additional value of this Figure to Table 2?

Conclusions

The discussion is relatively brief and could elaborate more on the findings.

The authors argue that in the elderly individuals of this study environmental factors like life style

might decrease heritability estimates. Yet, MRI studies in elderly twins show high heritability estimates compared to children and adults (see for overview Jansen et al. 2015, Neuropsychol Review). Moreover, as the authors already argue: heritability estimates based on genetic data are in general substantially lower than family or twin based estimates. Apart from array limitations, non-additive genetic factors are not taken into account when using genetic data. Would the authors expect those factors for brain measures?

With in addition: which genetic factors could explain the heritability that is observed, and how structure specific are these factors (highlighting the potential added value of genetic correlations in this study)?

The last paragraph on data reduction comes 'out of the blue' and does not really add important insights to the discussion.

Improvements:

In order to evaluate how specific the heritability estimates and related genetic factors are, in my opinion genetic correlations within and between the 7 different subcortical structures would complement this study. The genetic data are there. Have the authors considered these analyses, and if not, why?

Writing in general good, some details:

page 4, end row 56: add 'is'

page 12, row 225: instead of 'see and found' demonstrate/show

In conclusion: a very interesting, novel, and carefully conducted study, thanks for the ability to review.

REVIEWERS' COMMENTS:

Reviewer #1 (Remarks to the Author):

This is a strong revision of the manuscript. A good agreement between regional irritability patterns calculated in unrelated and related individuals greatly improves the confidence in the findings.

Minor note.

Please provide the link for the solar software (www.solar-eclipse-genetics.org)

Reviewer #2 (Remarks to the Author):

Thanks to authors for making the effort to address all my comments. I am satisfied with the revision and feel the manuscript is now suitable for publication.

Nature Communication

Ms. No.: NCOMMS-16-08591

Title: Heritability of the shape of subcortical brain structures in the general population.

We would like to thank the reviewers for careful consideration of our manuscript. Below, we provide a response to their comments and suggestions.

Reviewer #1

The manuscript by Roshchupkin and colleagues provides the first estimate of the heritability of the shape of subcortical structures. The volumes of subcortical structures are known to be highly heritable and therefore one would expect the high heritability of the shape as well. However, this manuscript shows that the regional pattern of heritability differs quite a bit and may not be related to the methodological error.

1. This manuscript is highly novel. The manuscript uses new methodological approaches for both shape and genetic analyses. This may also raise the criticism since the variance in regional heritability for the structures remains unexplained. This manuscript could be greatly strengthened if authors perform a replication study in the related individual. A dataset collected by human connectome project offers the same spatial resolution and heritability. It is collected in the related individuals and therefore heritability measurements could be calculated directly. I suggest repeating these analyses in the HCP dataset to show that the regional variance in heritability are indeed present and independent of the cohort or methodological issues.

We agree with the reviewer that replication of our findings in a different cohort can inform on the external validity of the results. Therefore, we have performed a replication analysis in the Queensland Twin Imaging Project (QTIM), a study of healthy young adult twins (n=1040). As we now describe in the **Results** section, we found that the regional variance in heritability is present despite differences in the populations, study design, scanner types, and methods to calculate the heritability (**Figure 1** here and **Supplementary Figure 4** in the manuscript). Yet, there was a significant correlation between the population-based and twin-based heritability estimates (Pearson's correlation coefficient = 0.28, $p = 3.03 \times 10^{-306}$).

Figure 1 Heritability maps of shape measures of subcortical brain regions under various models for QTIM.

We have made respective changes in **Methods**, **Results** and **Discussion** sections.

Reviewer #2

This study is the first to show that individual differences in the shape of subcortical brain structures is to an important extent heritable.

The study is novel and original in the field of brain studies by focusing on shape [instead of volume, white matter etc], and by using genetic data of unrelated individuals [instead of twin or family data].

Analyses are well performed, and additional analyses on reliability and data reduction provide added value to the interpretation and robustness of the results.

Yet, I have some questions related to the study.

Data

1. *The sample used is the Rotterdam Study that aims to study 'causes and determinants of chronic diseases in elderly' which raises the question whether the prevalence of disease is higher compared to the general population, and if yes, whether this would affect the results.*

We thank the reviewer for this question. While the Rotterdam Study was indeed set up with the aim to study 'causes and determinants of chronic diseases in elderly', the participants were not selected for the presence of diseases or risk factors. In fact, the main strength of the Rotterdam Study is its population-based setting and that it is representative of the general community-dwelling population. The overall response at baseline was 72.0 % (14,926 participants out of 20,744 invitees)¹, which is high for such an epidemiological study.

We have now clarified that the Rotterdam Study is a sample that is representative of the general population on page 5.

2. In addition the inclusion of subjects raises some questions:

On which basis were subjects included for scanning and genotyping (~35% of total sample)?

In principle all subjects were included for both scanning and genotyping. However, MRI scanning was introduced as part of the core protocol in 2005 with the purchase of a dedicated MRI scanner, while the Rotterdam Study was already initiated in 1990. All persons alive at that point were thus invited for a brain MRI, with currently about 13,000 scans performed in almost 6000 individuals. Genotyping was performed for all persons from whom blood was obtained and there was sufficient amount of DNA available (11,496 out of 14,926).

The **Methods** section now describes this in more detail.

3. After genetic QC, 3868 subjects remained in the study (page 7, but Table 1 states 3686?). After imaging QC no subjects were excluded?

We sincerely apologize for this typo and thank the reviewer for noticing this inconsistency. The final study population is 3686, as is stated in **Table 1**.

This is the number of subjects which were included in study after imaging and genetic QC. As we mentioned in “Study population” section the total number of unique subjects with MRI scanners were 5691, from which 75 were excluded based on imaging QC, resulting to 4774 subjects with information available on both genome-wide genotyping and MRI data. The rest were excluded based on pair-wise genetic relationship.

4. Statistics

The analyses and results are well presented, apart from some of the Figures that need clarification: I miss Figure Legends in Supplementary Figures 1 and 2. E.g., text x-axis is too small, and thus hard to read. In addition, Figure 4 is hard to read for me. What is meant with 'measure number' on the x-axis? What is the additional value of this Figure to Table 2?

We agree with the reviewer that some figures were not very clear. Therefore, we have changed the **Supplementary Figures 1-2** and added legends to them. Additionally, we edited **Figure 4**. The x-axis title ‘measure number’ referred to either a vertex or a principal component. Since this may be ambiguous, we have now replaced it with “vertex/component”. Regarding its additional value, unlike **Table 2**, which only describes the

maximal heritability, **Figure 4** shows the full range of vertex measures and components. While the maximal heritability is slightly lower for the components (**Table 2**), this additional figure indeed shows that this is consistently so for nearly all vertices, not only the top one.

5. Conclusions

The discussion is relatively brief and could elaborate more on the findings.

The authors argue that in the elderly individuals of this study environmental factors like life style might decrease heritability estimates. Yet, MRI studies in elderly twins show high heritability estimates compared to children and adults (see for overview Jansen et al. 2015, Neuropsychol Review). Moreover, as the authors already argue: heritability estimates based on genetic data are in general substantially lower than family or twin based estimates. Apart from array limitations, non-additive genetic factors are not taken into account when using genetic data. Would the authors expect those factors for brain measures?

We thank the reviewer for this comment. In response to point 1 of reviewer 1, we have now performed a replication analysis in QTIM, a study of healthy young adult twins. We found that the twin-based heritability tends to be higher than the population-based heritability. As the reviewer mentioned, this is expected based on the differences in the heritability calculation using GCTA (for the population-based sample) and SOLAR (for twins sample). This therefore suggests that such factors are also relevant for brain measures. We have now addressed this more explicitly in the **Discussion** on page 17.

6. *With in addition: which genetic factors could explain the heritability that is observed, and how structure specific are these factors (highlighting the potential added value of genetic correlations in this study)?*

This is indeed an important question. While heritability provides an estimate of how much of the variance is determined by genetics, it does not point to specific genetic loci. As such, our manuscript does not address that question. Instead, for gene discovery, the most commonly accepted method is to perform an unbiased screen of all genetic variants ('genome-wide association study') in order to identify specific genetic factors. However, such efforts require large-scale collaborations in the order of tens of thousands of individuals in order to identify a robust association²⁻⁴ Furthermore, additional multiple testing correction should be considered when performing GWAS of 54,000 shape measures. This could lead to a loss of power if the effects are homogeneous across a structure. However, if the effects are localized and mostly affect specific vertices, then a GWAS of shape measures may actually increase power since the effect sized will be larger compared to a GWAS of an aggregate volume.

We have expanded the **Discussion** with these considerations on page 13.

7. *The last paragraph on data reduction comes 'out of the blue' and does not really add important insights to the discussion.*

We partly agree with the reviewer. The reason to perform data reduction stems from the multiple testing burden (i.e. 54,000 shape measures). To reduce this burden, we performed a

principal component analysis on the correlated shape variables. Applying dimensionality reduction methods as an approximation to high-dimensional data analysis is currently a hot topic in the field, because classical methods of analyzing such large data sets are not feasible due to computational issues. As we showed in our principle component analysis it can lead to losing important information. Therefore, given the availability of new algorithms mentioned in manuscript, we think this is important message for the readers to take into account these results while planning their analysis.

8. *Improvements*

In order to evaluate how specific the heritability estimates and related genetic factors are, in my opinion genetic correlations within and between the 7 different subcortical structures would complement this study. The genetic data are there. Have the authors considered these analyses, and if not, why?

This is an excellent suggestion, which we have indeed considered ourselves. Briefly, the genetic correlation analyses could provide insight into whether different genetic factors might underlie the regional variance in heritability. However, we decided not to include genetic correlation analyses for several reasons. Importantly, we are very underpowered for such analyses, especially in our population-based sample. It will be difficult to establish any correlation slightly different than the extreme values (-1/0/1), making it impossible to determine nuanced differences. For corroboration of this, we refer the reviewer to the online power calculator for genetic correlation analyses:

<https://cnsgenomics.shinyapps.io/gctaPower/>

Furthermore, the computational requirements for performing all pairwise correlations between 54,000 vertices, i.e. about 3 billion tests, is tremendous. While heritability analyses on such a massive scale have been made possible, this remains unfeasible for genetic correlations (years of computation, even on a reasonably sized computer cluster). We also considered sampling random subsets of vertices or creating clusters based on the heritability maps, but these come with their own limitations.

For these and other reasons, we decided that a dedicated effort on genetic correlations is more appropriate to investigate this properly.

References

1. Hofman, A. *et al.* The Rotterdam Study: 2016 objectives and design update. *Eur. J. Epidemiol.* **30**, 661–708 (2015).
2. Bis, J. C. *et al.* Common variants at 12q14 and 12q24 are associated with hippocampal volume. *Nat. Genet.* **44**, 545–551 (2012).
3. Stein, J. L. *et al.* Identification of common variants associated with human hippocampal and intracranial volumes. **44**, (2012).
4. Hibar, D. P. *et al.* Common genetic variants influence human subcortical brain structures. *Nature* **8**, (2015).